# Normalization vs. Crops: Learning Gaze Representations via Constrained Rotation Optimization

## Abstract

Recent advances in appearance-based gaze estimation have adopted deep learning models to directly map face images to 3D gaze directions, but most existing methods rely on face normalization processes, which are costly and error-prone in unconstrained environments. While normalization-free approaches have been explored to address these challenges, they either discard the advantages of normalization in reducing appearance variability or lack a systematic understanding of the transformations involved. We revisit this problem and formalize crop-based gaze estimation through Constrained Rotation Optimization (CROp), which models face cropping as a virtual camera rotation and defines a consistent mapping between crop and camera coordinates. We further adopt multi-task learning to jointly estimate gaze and head pose, improving robustness without requiring explicit landmark-based preprocessing. Through extensive evaluation, we show that crop-based estimation, when treated rigorously, is a reliable alternative to normalization, especially under extreme head poses and noisy preprocessing. Our analysis highlights the trade-offs between the two approaches and offers practical guidelines for effective and robust gaze estimation in real-world, unconstrained settings.

## 1 Introduction

Eye gaze is an important non-verbal cue that communicates information about human attention and intent. It plays a central role in numerous applications, such as human-computer interaction (Zhang et al., 2019) and virtual/augmented reality (Sitzmann et al., 2018). For these reasons, achieving accurate gaze estimation is a relevant task in computer vision. Appearance-based gaze estimation methods have gained popularity in recent years due to their ability to map an eye or full-face image directly to their gaze direction in 3D space. Deep-learning models, such as convolutional neural networks (CNNs), can successfully learn this mapping from low-resolution images obtained using consumer-grade cameras, removing the need for specialized and expensive eye-tracking hardware.

Gaze estimation predicts a 3D vector in the camera coordinate system, but the high degree of freedom makes it difficult to learn the full range of appearance and gaze variations. To address this, a pre-processing step known as normalization or rectification (Sugano et al., 2014; Zhang et al., 2018) is widely adopted. Normalization confines the image space and reduces appearance variability by standardizing head pose and scale, while preserving a well-defined inverse mapping to the original camera coordinate system, interpretable as a virtual camera rotation between the original image and the normalized image. Many prior works have been developed upon this task formulation (Zhang et al., 2020; Cheng & Lu, 2022; Yin et al., 2024b; Bao & Lu, 2024; Yin et al., 2024a).

However, normalization faces significant challenges: its performance depends on accurate facial landmark detection and head pose estimation, which may be costly to compute and are often unreliable in unconstrained environments. Alternative strategies that bypass explicit normalization have been explored. One category estimates gaze in specialized coordinate systems that model the relative positions of face and camera (Kellnhofer et al., 2019; Zhang et al., 2022), but the properties and limitations of these representations remain underexplored. Another approach directly regresses the 3D gaze vector in the camera coordinate system from full images (Balim et al., 2023). How-

Figure 1: Overview of our gaze estimation framework. Leveraging our Constrained Rotation Optimization mechanism, we estimate 3D gaze directly from face crops, training a multi-task gaze model that simultaneously regresses head pose information.

ever, direct approaches introduce greater appearance variability, hindering generalization to novel environments.

In this work, we formalize a normalization-free formulation of gaze estimation through a novel Constrained Rotation Optimization (CROp), which interprets face cropping as a virtual camera rotation in 3D space (Fig. 1). CROp establishes a mathematically precise mapping between the crop and camera coordinate systems, enabling accurate conversion of gaze vectors between them. This preserves the geometric consistency of normalization while removing its reliance on landmarks and explicit pose estimation. To compensate for the missing head pose input, we adopt a multi-task learning framework that jointly estimates gaze and head orientation, using recent differentiable rotation representations (Levinson et al., 2020; Roth & Gavrila, 2023).

We conduct a systematic comparison between normalization and crop-based gaze estimation. Across multiple datasets, head pose distributions, and pre-processing conditions, we show that with a precise geometric transformation like CROp, learning from unnormalized inputs is not only feasible but also robust. CROp achieves superior performance in challenging scenarios such as extreme head poses or noisy pre-processing, while offering a simpler and more efficient pipeline. These findings validate our approach and provide practical insights for deploying gaze estimation systems in unconstrained environments.

## 2 RELATED WORKS

### 2.1 APPEARANCE-BASED GAZE ESTIMATION

Appearance-based gaze estimation has emerged as a prominent approach due to its ability to operate under unconstrained conditions with minimal hardware demands (Zhang et al., 2015; 2021; Ghosh et al., 2023; Cheng et al., 2024). Foundational work on data normalization (Sugano et al., 2014; Zhang et al., 2018) established essential strategies for handling variations in head pose and camera geometry. The introduction of large-scale datasets (Zhang et al., 2017a; Krafka et al., 2016; Funes Mora et al., 2014; Zhang et al., 2020) has enabled the training of more robust and generalized models that maintain high accuracy across different environments. Recent efforts have continued to refine network architectures and training protocols (Zhang et al., 2020; Cheng & Lu, 2022; Cheng et al., 2022; Xu et al., 2023; Yin et al., 2024b; Bao & Lu, 2024; Yin et al., 2024a; Qin et al., 2025).

On the other hand, several works have explored gaze estimation without relying on explicit normalization. One approach is to directly map the face image to 2D coordinates on the screen (Huang et al., 2017; Krafka et al., 2016). However, these methods are constrained to fixed screen-based settings and do not generalize well to different camera viewpoints or arbitrary 3D gaze estimation. Other works explore an approach that outputs 3D gaze vectors from face images without normalization. Gaze360 (Kellnhofer et al., 2019) estimates gaze in an eye coordinate system defined from the center of the face. Their method assumes an upright orientation with no roll, while our approach is based on minimizing the projection error of crop points. Although this representation has been used as an alternative to normalization, a systematic study of its impact and geometric properties is missing. GazeOnce (Zhang et al., 2022) adopts a multi-task learning framework that jointly estimates gaze direction along with auxiliary tasks such as face localization. However, they do not explicitly

estimate the 3D gaze vector in the camera coordinate system, making it difficult to determine the PoG in real-world scenarios. EFE (Balim et al., 2023) directly regresses the 3D gaze vector in the camera coordinate system using the entire camera image as input. While this avoids pre-processing, the wide appearance variability increases task complexity, and its generalization performance in unseen environments remains uncertain. In contrast, we propose CROp that replaces the normalization process by modeling face cropping as a geometric transformation. This approach preserves the benefits of appearance-space constraints while eliminating the need for precise face alignment.

## 2.2 HEAD POSE ESTIMATION

Head pose estimation (HPE) is a widely studied task in computer vision due to its broad applications, including human-computer interaction and human behaviour analysis. Most existing approaches focus on the 3 Degrees-of-Freedom (DoF) head rotation (yaw, pitch, and roll) (Hempel et al., 2024; Li et al., 2022; Huang et al., 2020), while some aim to estimate a 6 DoF pose that includes the translation (Algabri et al., 2024; Albiero et al., 2021; Roth & Gavrila, 2023).

Traditional methods have adopted the landmark-based approach (Werner et al., 2017; Gupta et al., 2019) that localizes facial keypoints as a first step, and subsequently aligns a 3D head model through Perspective-n-Point (Huang et al., 1995) algorithms. This approach, commonly used in gaze estimation pipelines, can produce highly accurate results in controlled environments but is sensitive to errors in landmark estimation (Chang et al., 2017). To address these issues, research has shifted towards landmark-free approaches that directly regress head pose from images (Ruiz et al., 2018; Hempel et al., 2022; Ahn et al., 2014). A key challenge for these methods is the choice of the rotation representation. Euler angles are intuitive but discontinuous and suffer from gimbal lock at large rotations (Zhou et al., 2019). Some works have used alternative representations such as quaternions (Hsu et al., 2018) and 6D representations (Hempel et al., 2022) to improve stability. Recently, a 9D continuous and differentiable representation named SVDO$^+$ was proposed (Levinson et al., 2020) and first applied to head pose estimation in Roth & Gavrila (2023), demonstrating its effectiveness. We also adopt SVDO$^+$ to enable joint regression of gaze and 3 DoF head pose.

Similar to gaze estimation, when estimating head pose from a face image, it is not obvious how to revert it to the original camera coordinate system. Several works have explored predicting head pose from face crops while handling transformations to the original camera space. Img2Pose (Albiero et al., 2021) formulates head pose estimation within an object detection framework, and directly regresses pose from crop proposals. IntrApose (Roth & Gavrila, 2023) further improves this by incorporating camera intrinsics. Li et al. (2022) defines a virtual camera rotation to rectify face images before performing head pose estimation. These methods highlight the importance of handling coordinate transformations when working with cropped images. Inspired by this, our approach extends the same idea to gaze estimation, ensuring accurate gaze recovery without requiring explicit normalization.

## 2.3 GAZE AND HEAD POSE JOINT ESTIMATION

In addition to appearance-based approaches focusing solely on the eyes or face region, some methods explicitly integrate head and body pose information to handle more dynamic and unconstrained scenarios. Zhu & Deng (2017) introduced a two-step approach that first extracts features via a CNN, then applies a parametric geometry-based model to compute 3D gaze angles. Nonaka et al. (2022) proposed a method for dynamic 3D gaze estimation from a distance by modeling temporal eye-head-body coordination, enabling robust performance in real-world settings. Kothari et al. (2021) proposed a weakly supervised approach with the look at each other (LAEO) loss, which depends on 3D head pose estimation. These works underline the significance of jointly leveraging head and body pose cues to enhance gaze estimation accuracy and robustness across diverse environments. Similarly, we propose a framework that jointly estimates gaze and head pose, addressing the challenge of unknown head rotation in cropped images without relying on facial landmark alignment.

## 3 METHOD

We present a gaze estimation formulation that avoids face alignment in the pre-processing pipeline. By establishing a precise mathematical relationship between the original camera coordinate system

and the cropped face region, our approach provides a solid foundation for accurate gaze estimation from raw face crops.

## 3.1 OVERVIEW

Extracting a face crop from an image can be interpreted as defining a new *virtual camera* whose optical axis is centered on the crop, resulting in a transformed perspective from the original camera. However, cropping does not exactly correspond to a rigid camera transformation, and no single rotation can perfectly align the two views. To resolve this, we formulate a constrained optimization problem that defines the best-approximating rotation by minimizing the projection error of a set of anchor points (the crop center and corners). The resulting rotation matrix provides a well-defined transformation between crop and camera coordinates, allowing models to be trained directly on face crops while still enabling precise conversion between the two systems.

Unlike normalization pipelines, which explicitly estimate head pose during pre-processing, our crop-based formulation starts without head pose information. To complement this, we adopt a multi-task design where the model jointly predicts gaze and head orientation. This serves two purposes: it improves gaze accuracy by leveraging the natural coupling between head and eye movements, and it produces head pose estimates useful for downstream applications without additional pre-processing.

## 3.2 CONSTRAINED ROTATION OPTIMIZATION

To formalize our approach mathematically, we establish the relationship between the original camera coordinate system and that of the face crop. We define the crop bounding box using the top-left $(x_1, y_1)$ and bottom-right $(x_2, y_2)$ corners. The camera intrinsic matrices of the original image $K_{\text{cam}}$ and the one for the crop camera $K_{\text{crop}}$ are defined as:

$$K_{\text{cam}} = \begin{bmatrix} f_x & 0 & c_x \\ 0 & f_y & c_y \\ 0 & 0 & 1 \end{bmatrix} \quad (1) \qquad K_{\text{crop}} = \begin{bmatrix} f_x & 0 & (x_1+x_2)/2 \\ 0 & f_y & (y_1+y_2)/2 \\ 0 & 0 & 1 \end{bmatrix} \quad (2)$$

Here, $f_x$ and $f_y$ are the focal lengths, $(c_x, c_y)$ is the principal point in the full image, while the principal point in the cropped image coincides with the crop's center. Since the crop is a cut-out of the original image, its pixel coordinates differ only by translation.

In the standard pinhole camera model, the camera matrix $K_{\text{cam}}$ projects a point in 3D camera coordinates, $p$, onto the image plane as $u = [x_u \cdot s, y_u \cdot s, s]^\top = K_{\text{cam}} \cdot p$, where $s$ is a non-zero scale factor. Conversely, given an image point expressed in homogeneous coordinates, $u = [x_u, y_u, 1]^\top$, we can compute the ray intersecting it in camera coordinates (represented by a unit vector) by applying the inverse of the camera matrix $K_{\text{cam}}$:

$$\hat{p} = \frac{K_{\text{cam}}^{-1} u}{||K_{\text{cam}}^{-1} u||}. \qquad (3)$$

The same operation applies to a point or vector in crop space, using $K_{\text{crop}}$ instead of $K_{\text{cam}}$.

We consider the center point of the bounding box $c = [(x_1+x_2)/2, (y_1+y_2)/2, 1]^\top$, and the matrix consisting of stacked four corner points $P \in \mathbb{R}^{3 \times 4}$ defined as

$$P = \begin{bmatrix} x_1 & x_2 & x_1 & x_2 \\ y_1 & y_1 & y_2 & y_2 \\ 1 & 1 & 1 & 1 \end{bmatrix}. \qquad (4)$$

Using Eq. (3), we compute the rays intersecting these points in camera coordinates, $\hat{c}_{\text{cam}}, \hat{P}_{\text{cam}}$, and the corresponding rays in crop coordinates, $\hat{c}_{\text{crop}}, \hat{P}_{\text{crop}}$ using $K_{\text{crop}}$ instead of $K_{\text{cam}}$. These rays represent the 3D directions from the respective camera origins through each point, providing the geometric relationship between the two coordinate systems.

Ideally, the desired coordinate transformation aligns the vectors in camera coordinates to the matching ones in crop coordinates, *i.e.*, a rotation $R$ such that $[\hat{c}_{\text{crop}}, \hat{P}_{\text{crop}}] = R \cdot [\hat{c}_{\text{cam}}, \hat{P}_{\text{cam}}]$. However, since a face crop is not a perfect rigid transformation in 3D space, this exact alignment is generally impossible to achieve. Therefore, we formulate an approximate solution.

We prioritize the alignment of the camera's optical axis ($z$-axis) to the center of the crop as our primary constraint, ensuring that the virtual crop camera is directly facing the center of the face. Then, we seek the rotation $\boldsymbol{R}^*$ that optimally aligns the corner vectors, solving the constrained optimization problem:

$$\min_{\boldsymbol{R} \in SO(3)} \quad \frac{1}{2}\|\hat{\boldsymbol{P}}_{\text{crop}} - \boldsymbol{R} \cdot \hat{\boldsymbol{P}}_{\text{cam}}\|_F^2 \tag{P}$$
$$\text{s.t.} \quad \hat{\boldsymbol{c}}_{\text{crop}} = \boldsymbol{R} \cdot \hat{\boldsymbol{c}}_{\text{cam}}.$$

In practice, $\boldsymbol{R}^*$ can be efficiently computed by finding the rotation that aligns $\hat{\boldsymbol{c}}_{\text{cam}}$ to $\hat{\boldsymbol{c}}_{\text{crop}}$ and then finding the rotation about $\hat{\boldsymbol{c}}_{\text{crop}}$ that optimally aligns the corner points. This can be solved using the Kabsch algorithm (Kabsch, 1976; 1978; Umeyama, 1991), which finds the optimal rotation between two sets of points. This operation is the core of our CROp approach.

To summarize, CROp computes $\boldsymbol{R}^*$ when given the crop bounding box and the camera intrinsic matrix $\boldsymbol{K}_{\text{cam}}$. Then, a gaze vector $\boldsymbol{g}_{\text{cam}}$ expressed in camera coordinates (*e.g.*, a label for a training sample) is converted to crop coordinates as $\boldsymbol{g}_{\text{crop}} = \boldsymbol{R}^* \cdot \boldsymbol{g}_{\text{cam}}$, while a gaze vector predicted in crop coordinates can be converted to original camera coordinates as $\boldsymbol{g}_{\text{cam}} = \boldsymbol{R}^{*-1} \cdot \boldsymbol{g}_{\text{crop}}$.

### 3.3 MULTI-TASK GAZE MODEL

Using CROp, we can estimate the gaze from simple face crops, avoiding the normalization procedure and thus ignoring head pose. To recover head pose within our pipeline, we adopt a multi-task model that learns it jointly with gaze. In detail, we repurpose our gaze model into one that predicts, from an input face image, both the gaze and the 3 DoF head pose (rotation). The predicted gaze direction $\hat{\boldsymbol{g}}$ is represented by the 2D vector containing its pitch and yaw angles and converted to a 3D vector through differentiable operations. For gaze estimation, we use the angular loss:

$$\mathcal{L}_{\text{angular}} = \arccos\left(\frac{\hat{\boldsymbol{g}} \cdot \boldsymbol{g}}{\|\hat{\boldsymbol{g}}\|\|\boldsymbol{g}\|}\right). \tag{5}$$

For head pose, we adopt SVDO$^+$ (Levinson et al., 2020), a 9D continuous and differentiable representation for rotations. This representation avoids the discontinuities and gimbal lock issues associated with Euler angles, making it particularly suitable for deep learning-based rotation regression. Calling $\hat{\boldsymbol{H}}$ the predicted head pose, we use the geodesic loss:

$$\mathcal{L}_{\text{geodesic}} = \arccos\left(\frac{\text{tr}(\boldsymbol{H}\hat{\boldsymbol{H}}^\top) - 1}{2}\right). \tag{6}$$

Our model is trained to optimize the multi-task loss

$$\mathcal{L}_{\text{tot}} = \mathcal{L}_{\text{angular}} + \mathcal{L}_{\text{geodesic}}. \tag{7}$$

The head pose is predicted in the same crop coordinate system as the gaze and can be converted to camera coordinates using the transformation determined by CROp. Specifically, the head pose in camera coordinates is obtained as $\hat{\boldsymbol{H}}_{\text{cam}} = \boldsymbol{R}^*\hat{\boldsymbol{H}}$.

## 4 EXPERIMENTS

We empirically study the effectiveness of CROp for learning gaze directly from face crops as an alternative to normalization in both cross-domain and within-domain settings. We then ablate the components of our framework and demonstrate its robustness under strong and artificial degradation.

### 4.1 EXPERIMENTAL SETTINGS

**Gaze Datasets.** We use four different gaze estimation datasets that provide the 3D eye gaze in camera coordinates, as well as accurate camera intrinsics and annotated head pose, to allow the comparison with normalization. **MPIIFaceGaze** (Zhang et al., 2017b) (MPII) contains 15 subjects

looking at on-screen targets on their laptops under various lighting conditions, both indoor and outdoor. **ETH-XGaze** (Zhang et al., 2020) (XG) is a large dataset collected using 18 high-resolution cameras in a studio environment. We use only the train set with publicly available annotations, comprising 80 subjects and 756K images. **EYEDIAP** (Funes Mora et al., 2014) (ED) is a video dataset that contains 16 subjects gazing at continuous screen targets and 3D floating objects. **EVE** (Park et al., 2020) is a video dataset containing 54 participants recorded while looking at on-screen stimuli. We extract frames from the videos with a stride of 10 for more efficient training.

**Model Architecture.** Both our CROp formulation and the proposed multi-task loss are model-agnostic and applicable to any backbone network for gaze estimation. We conduct our main experiments using the ResNet18 (He et al., 2016) backbone, which has been shown in prior work to be highly competitive with larger networks for gaze estimation (Zhang et al., 2022; Liu et al., 2021). In Appendix A, we further demonstrate the versatility of our approach across different architectures.

**Data Pre-processing.** For normalization (Norm), we follow the standard procedure with focal length 960, camera distance 300, and a $448 \times 448$ ROI, ensuring consistency across datasets. Gaze estimation benchmarks typically apply normalization using manually annotated facial landmarks. To cover real-world scenarios, where ground truth annotations are not available, we evaluate in two settings. In the ideal setting (**GT**), Norm relies on ground-truth landmarks, while CROp is provided with ideal bounding boxes centered on the face and with size proportional to the maximum keypoint distance (scaled by 1.5, or 2 in EYEDIAP, where only two eye landmarks are available). In the realistic setting (**Det**), we use InsightFace (Guo & Deng, 2018). Norm employs detected bounding boxes and 106 facial landmarks, followed by head pose estimation via the PnP algorithm. CROp instead uses only the detected bounding boxes and does not require landmarks.

**Implementation Details.** The inputs of gaze models are resized to $256 \times 256$. Models are trained for 25 epochs using the Adam optimizer (Kingma & Ba, 2015), with a learning rate of $1 \times 10^{-4}$ decayed by 0.1 every 10 epochs.

### 4.2 CROp vs Normalization

We first evaluate whether our gaze estimation scheme, comprising CROp and joint head-pose learning, enables more accurate predictions than the standard normalization approach. We conduct cross-dataset and within-dataset evaluations, as well as error distribution analysis.

#### 4.2.1 Cross-Dataset Evaluation

We compare gaze models trained using the standard normalization scheme with our CROp approach in Table 1. Our approach achieves lower angular errors in most cases, demonstrating that gaze estimation can be effectively performed without normalization. Notably, the improvement is especially consistent on XGaze, where our method achieves an average error reduction of 6.52%, and on EYEDIAP, with a reduction of 10.61%, under ideal pre-processing (GT).

Our results in Table 1 also confirm that crop-based gaze estimation remains robust under realistic pre-processing (Det), achieving the best performance in most cases when using automated face detection. Notably, using a face detector does not significantly degrade accuracy in cross-dataset experiments, likely being overshadowed by stronger sources of domain shift. Our method proves particularly robust on MPIIFaceGaze, where natural lighting and lower-quality laptop webcams can make normalization more susceptible to noise. These findings highlight the adaptability of our formulation, demonstrating its reliability without the need for strict pre-processing procedures.

To gain insight into the advantages of crop-based estimation, we analyze the error distribution in a particularly challenging case where it achieves strong performance: MPIIFaceGaze → XGaze. Although the average errors are large for both methods, our approach consistently outperforms Norm on a per-subject basis ($p = ***$, Wilcoxon signed-rank test (Woolson, 2005)). Figure 3 shows how errors vary across head pose angles (pitch, yaw, and roll), computed in the original camera coordinate system, which is common to both methods. The crop-based representation tends to reduce errors across all head poses, but finds the largest relative improvements at extreme head angles. Although normalization is designed to mitigate head pose variation, we observe that it still struggles under extreme poses. Figure 2 illustrates this effect: for moderate head poses, our face-crop approach resembles normalized images, while at wider angles the two representations diverge significantly,

Table 1: Cross-dataset evaluation results of our approach (CROp and multi-task loss) versus normalization. We report the results under both pre-processing from annotations (GT) and a face detector (Det). All values are angular errors (°).

| Train | Method | Pre-proc. | MPII | XG | EVE | ED |
|-------|--------|-----------|------|------|------|------|
| MPII | Norm | GT | - | 31.23 | 11.10 | 17.68 |
| | Ours | GT | - | **29.37** | **10.19** | **15.62** |
| | Norm | Det | - | 30.55 | **11.43** | 17.46 |
| | Ours | Det | - | **29.56** | 11.44 | **15.79** |
| XG | Norm | GT | **7.19** | - | 9.76 | **12.15** |
| | Ours | GT | 7.41 | - | **8.49** | 12.81 |
| | Norm | Det | 8.76 | - | 9.46 | 13.19 |
| | Ours | Det | **7.76** | - | **8.98** | **12.84** |
| EVE | Norm | GT | 9.69 | 39.71 | - | 21.32 |
| | Ours | GT | **8.79** | **36.90** | - | **19.32** |
| | Norm | Det | 9.53 | 39.13 | - | 20.47 |
| | Ours | Det | **8.51** | **36.01** | - | **18.96** |

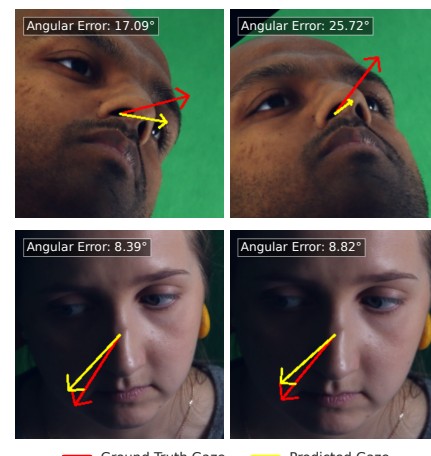

Figure 2: Qualitative comparison of CROp using face crops (left) and normalization (right), on XGaze.

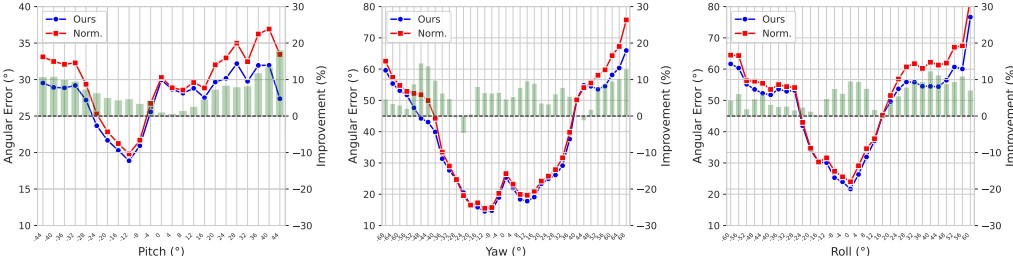

Figure 3: Angular error as a function of head pose (pitch, yaw, roll) using CROp versus the normalization-based approach. The plots show absolute errors along with the relative improvement.

but in both cases the resulting faces remain far from a canonical upright view. In these cases, direct estimation from face crops proves more reliable, suggesting that such representations can be advantageous in unconstrained conditions.

### 4.2.2 WITHIN-DATASET EVALUATION

We further conduct within-dataset experiments on MPIIFaceGaze, XGaze, and EVE. For MPI-IFaceGaze, we follow the standard 15-fold cross-validation protocol. To align with prior work, we use a $448 \times 448$ input size for this dataset. On XGaze, we train on the first 60 subjects and test on the remaining 20. For EVE, we report results on both the validation set (using publicly available annotations) and the test set (via the official server). Unlike in cross-dataset settings, where multiple factors contribute to domain shift, within-dataset experiments isolate pre-processing noise as the primary source of variation. To mitigate its impact, we incorporate bounding box augmentations during training, applying rescaling and translations up to 20% of the bounding box size. Table 2 reports the results of this evaluation and shows that our approach consistently outperforms normalization, especially in the Det scenario, highlighting the robustness of crop-based gaze estimation under realistic conditions.

### 4.2.3 COMPUTATIONAL COST

In Table 3 we show the runtime latency of our pipeline compared to normalization, breaking down the cost of each component (averaged over the XGaze dataset). Our pipeline avoids entirely the landmark detector model, and saves 23% of the total computation time.

Table 2: Within-dataset evaluation results of our approach (CROp and multi-task loss) versus normalization.

| Method | Pre-proc. | MPII | XG | EVE Val | EVE Test |
|--------|-----------|------|------|---------|----------|
| Norm | GT | **4.93** | 5.30 | 4.50 | 4.99 |
| Ours | GT | 5.02 | **5.13** | **4.46** | **4.51** |
| Norm | Det | 5.32 | 6.45 | 4.71 | 5.10 |
| Ours | Det | **4.98** | **5.55** | **4.69** | **4.99** |

Table 3: Runtime comparison on RTX A6000. Numbers are ms.

| Method | Face Det. | Lmk Det. | Norm | CROp | Model | Total |
|--------|-----------|----------|------|------|-------|-------|
| Norm | 10.88 | 2.79 | 2.67 | - | 4.40 | 20.74 |
| Ours | 10.88 | - | - | 0.69 | 4.40 | 15.97 |

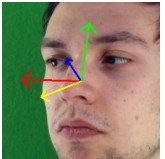 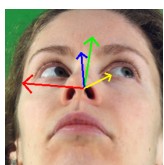 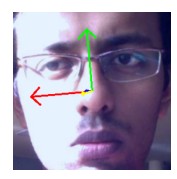 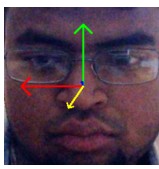 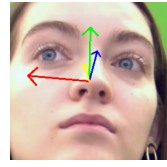 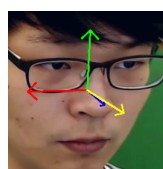

(a) MPIIFaceGaze →XGaze.  (b) XGaze →MPIIFaceGaze  (c) XGaze →EVE

Figure 4: Qualitative results of CROp estimating gaze direction (yellow) and head pose (RGB axes).

### 4.3 ABLATION STUDIES

In Table 4 we examine the two main components of our approach: the direct handling of crops using CROp, and the effectiveness of multi-task learning for gaze and head pose. To assess whether explicit head pose estimation benefits gaze due to their inherent coupling, we also include results for a multi-task model trained on normalized face images.

Using CROp without joint learning of head pose already provides an advantage over standard normalization in all scenarios. Adding head pose supervision further improves overall performance, allowing our full approach to achieve the best results in most cases. While there are a few cases where multi-task learning slightly reduces gaze estimation accuracy, this is likely due to the model's balancing between gaze and head pose estimation. However, this trade-off is offset by the benefit of obtaining head pose information directly, without additional computational steps. We also observe that multi-task loss benefits the normalization-based method as well. However, for normalization-based models, the head pose is already computed during pre-processing, making explicit estimation redundant. In contrast, our method efficiently recovers head pose directly from face crops while simultaneously improving gaze estimation without adding to computational cost (Fig. 4).

### 4.4 ADDITIONAL ANALYSES

#### BOUNDING BOX NOISE AND OCCLUSIONS

We have demonstrated CROp's adaptability to a real-world face analysis pipeline like InsightFace, without the need for manually curated face bounding boxes. To further evaluate its robustness, we conduct experiments with synthetic noise and occlusions, simulating additional degradation in the automated pre-processing. We focus on MPIIFaceGaze, as it is a visually challenging dataset representing natural scenarios, and analyze its behavior under extreme artificial degradation that mimics real-world pre-processing failures. We introduce noise by applying random jittering and rescaling to bounding boxes and simulate occlusions by masking face regions with randomly placed black boxes of varying sizes, affecting both methods.

As shown in Table 5, CROp remains robust to moderate bounding box noise, outperforming normalization-based estimation up to ±30% translation and rescaling. At higher noise levels, normalization benefits from the additional step of facial landmark localization, but beyond this point, its performance also starts to degrade, indicating failures in landmark detection. Table 6 presents results under increasing levels of occlusion. Since facial landmark detection is sensitive to occlusions, normalization suffers a significant performance drop as occlusion severity increases. Crop-based estimation, while also affected, degrades more smoothly and maintains lower absolute errors across all levels. These results suggest that crop-based estimation provides reliable performance under re-

Table 4: Impact of head pose supervision.

| Train | Method | HPE | MPII | XG | EVE | ED |
|-------|--------|-----|------|-------|-------|-------|
| MPII | Norm. | | - | 31.23 | 11.10 | 17.68 |
| | Norm. | ✓ | - | 30.27 | 10.84 | 17.66 |
| | CROp | | - | **28.29** | 11.03 | 16.88 |
| | CROp | ✓ | - | 29.37 | **10.19** | **15.63** |
| XG | Norm. | | 7.19 | - | 9.76 | 12.15 |
| | Norm. | ✓ | **6.94** | - | 9.42 | **9.70** |
| | CROp | | 7.17 | - | 8.86 | 11.91 |
| | CROp | ✓ | 7.41 | - | **8.49** | 12.81 |
| EVE | Norm. | | 9.69 | 39.71 | - | 21.32 |
| | Norm. | ✓ | 9.46 | 37.14 | - | 21.82 |
| | CROp | | 8.89 | 37.48 | - | 21.12 |
| | CROp | ✓ | **8.79** | **36.90** | - | **19.32** |

Table 5: Impact of bounding box noise. Random scaling ($\pm X\%$) and translation ($X\%$ of box size); models trained on EVE.

| Method | Bounding Box Noise | | | | | |
|--------|------|------|------|------|-------|-------|
| | 0% | 10% | 20% | 30% | 40% | 50% |
| Norm. | 9.53 | 9.53 | 9.53 | **9.54** | **9.56** | **9.63** |
| Ours | **8.51** | **8.66** | **8.92** | **9.54** | 10.25 | 10.83 |

Table 6: Impact of occlusions. Occlusions cover $X\%$ of bounding box area; models trained on EVE.

| Method | Occlusion | | | | | |
|--------|------|------|-------|-------|-------|-------|
| | 0% | 10% | 20% | 30% | 40% | 50% |
| Norm. | 9.53 | 9.74 | 10.13 | 10.48 | 10.86 | 11.17 |
| % Increase | - | 2.2% | 6.3% | 10.0% | 14.1% | 17.2% |
| Ours | **8.51** | **8.68** | **8.93** | **9.22** | **9.58** | **9.90** |
| % Increase | - | 2.0% | 4.9% | 8.3% | 12.6% | 16.3% |

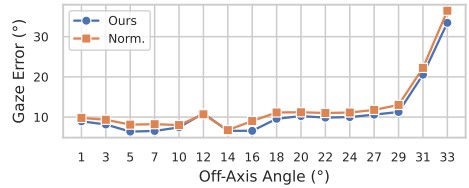

Figure 5: Error vs off-axis angle on EVE.

alistic amounts of noise and occlusion, while normalization can compensate for noise to some extent through additional preprocessing, but remains ultimately vulnerable to preprocessing failures.

PERSPECTIVE DISTORTION AND EXTREME POSES

A common challenge in gaze estimation is to handle perspective distortion, particularly when faces are close to the camera or viewed at large off-axis angles. Notably, normalization, which relies on a planar face assumption, also suffers in these conditions. We analyze this effect in Fig. 5 using the EVE dataset, which includes the widest viewing angles (up to $\sim 30°$) due to its left and right webcam views. While our method shows some degradation at large off-axis angles, where the approximation in $\boldsymbol{R}^*$ becomes less accurate, the same happens for normalization. In fact, considering only the left and right cameras in EVE, and using models trained on XGaze, our method achieves a lower average error of $10.01°$, compared to $11.19°$ for normalization.

## 5 CONCLUSION

We revisited pre-processing in appearance-based gaze estimation by comparing normalization and crop-based representations. To formalize the latter, we introduced CROp, a constrained optimization that models cropping as a virtual camera rotation and defines a consistent mapping between crop and camera coordinates. We further adopted a multi-task design that jointly predicts gaze and head pose, improving robustness while avoiding explicit landmark-based preprocessing. Our analysis across datasets and degradation scenarios shows that crop-based estimation is a reliable alternative to normalization if treated rigorously, particularly under realistic detector pipelines and at extreme head poses. Our results highlight the trade-offs between normalization-based and crop-based approaches and provide practical guidelines for deploying gaze estimation in real-world, unconstrained settings. While our method effectively estimates gaze direction, it assumes the gaze origin coincides with the bounding box center rather than the true anatomical origin point. Future work could explicitly model and estimate the precise gaze origin within the facial structure, yielding more anatomically correct gaze vectors and improving overall accuracy.

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

## A    DIFFERENT BACKBONES

Our main experiments evaluated CROp using a ResNet18 backbone (He et al., 2016). However, our approach is model-agnostic and compatible with any gaze estimation architecture. To validate this versatility, Table 7 presents results obtained when applying our method to two additional popular backbones for gaze estimation: ResNet50 and the transformer-based GazeTR-Hybrid (Cheng & Lu, 2022) based on ResNet18. The results are consistent with our earlier findings, as CROp achieves the best performance in the majority of cases and remains highly competitive with normalization. Normalization only maintains a slightly larger performance margin on the XGaze dataset. This may be attributed to the challenging distribution of head poses in XGaze, which can influence the learning dynamics of our multi-task model, steering it toward head pose estimation.

## B    HEAD POSE ESTIMATION ACCURACY

In our proposed gaze estimation pipeline, head pose is not explicitly computed during normalization. Instead, it is estimated jointly with gaze through a multi-task model that performs direct rotation regression from the input image. While our primary focus is on gaze estimation, we evaluate the accuracy of head pose estimation (3-DoF) to assess the effectiveness of this joint prediction. As a baseline, we consider a standard keypoint-based approach: facial landmarks are detected, and a

| Train | Method | MPII | XG | EVE | ED |
|-------|--------|------|-----|-----|-----|
| **ResNet50** | | | | | |
| MPII | Norm. | - | 34.45 | 14.47 | 19.75 |
|      | Ours  | - | **30.77** | **11.52** | **17.14** |
| XG   | Norm. | **6.30** | - | **7.19** | **9.21** |
|      | Ours  | 7.06 | - | 8.00 | 11.75 |
| EVE  | Norm. | 9.72 | 38.55 | - | 22.78 |
|      | Ours  | **8.44** | **35.66** | - | **19.92** |
| **GazeTR-Hybrid** | | | | | |
| MPII | Norm. | - | 29.46 | 12.15 | 19.94 |
|      | Ours  | - | **27.00** | **11.21** | **12.57** |
| XGaze | Norm. | **7.20** | - | **9.42** | **10.29** |
|       | Ours  | 8.44 | - | 9.87 | 14.58 |
| EVE  | Norm. | 12.01 | 40.75 | - | 22.39 |
|      | Ours  | **9.34** | **30.97** | - | **17.87** |

Table 7: Comparison of gaze estimation performance across different backbones (ResNet50 and GazeTR-Hybrid).

| Method | Train | MPII | XG | EVE | ED |
|--------|-------|------|-----|-----|-----|
| CROp | MPII | – | 41.19 | 29.47 | 18.88 |
|      | XG   | **14.12** | – | **14.51** | **13.70** |
|      | EVE  | **17.05** | 21.34 | – | **6.72** |
| InsightFace | – | 20.19 | 18.83 | 20.32 | 14.56 |

Table 8: Head pose estimation error for our multi-task method (CROp) compared to a keypoint-based baseline (InsightFace). Numbers are MAE of the geodesic distance (°).

generic 3D face model is fit to estimate head rotation. This method is commonly used in normalization pipelines. To ensure consistency with our other experiments, we use keypoints detected by InsightFace, which are also used to assess normalization in realistic detection scenarios (Det). Accordingly, our CROp-based models are evaluated on face crops obtained from the same InsightFace detections.

Table 8 reports the mean angular error (MAE) of the geodesic distance (Roth & Gavrila, 2023) between the predicted and the ground-truth head rotations. Since the keypoint-based baseline does not rely on training data, its performance remains fixed. In contrast, our multi-task regression model's accuracy varies depending on the training distribution. When trained on datasets with limited head pose variation (e.g., MPIIFaceGaze), direct regression underperforms. However, with large-scale datasets such as XGaze and EVE, which feature diverse head poses, our method consistently outperforms the keypoint-based baseline. This highlights the effectiveness of recovering head pose jointly with gaze in a unified, training-based framework

## C  PERSPECTIVE DISTORTION ON SYNTHETIC DATA

In Section 4.4 we analyzed how performance degrades for CROp and normalization when faces are viewed at large off-axis angles, leading to strong perspective distortion. The EVE dataset provided real-world examples of such scenarios, with viewing angles reaching up to approximately $30°$.

To further evaluate performance under more extreme conditions, we conduct additional experiments using synthetic data. Following previous work (Qin et al., 2024), we apply multi-view reconstruction to the ETH-XGaze dataset and synthesize images from arbitrary novel viewpoints. To obtain larger

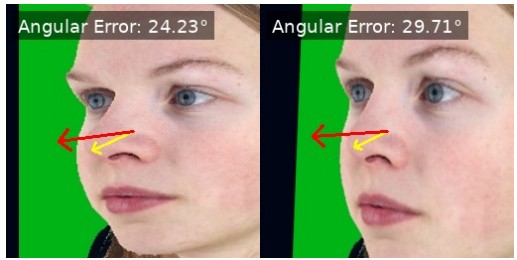

Figure 6: Comparison of CROp (left) and normalization (right) on synthetic data. Ground truth is in red, gaze prediction is in yellow.

| Train | Norm. | Ours |
|-------|-------|------|
| MPIIFG | 31.79 | **31.36** |
| EVE | 37.21 | **36.36** |

Table 9: Performance on synthetic XGaze. Numbers are angular errors (°).

| Train | MPIIFG | ED | Avg. |
|-------|--------|------|------|
| XG | 7.41 | **12.81** | 10.11 |
| EVE | 8.79 | 19.32 | 14.06 |
| XG + EVE | **6.36** | 13.36 | **9.86** |

Table 10: Results of CROp trained on multiple datasets. Numbers are angular errors (°)

off-axis angles, we increase the camera's field of view, shift the face away from the optical axis and closer to the camera. Specifically, we select the original frontal camera 0 and use the first five frames of each subject. For each selected image, we generate eight new views with varied camera positions, resulting in a total of 3,200 synthetic samples. This results in highly off-axis views of approximately $45°$, introducing significant perspective distortion in the resulting face crops for both our method and normalization (see Fig. 6). As shown in Table 9, our method continues to outperform normalization even under these challenging synthetic condition, demonstrating that gaze estimation from face crops is robust even under strong perspective distortion.

## D    MULTIPLE TRAINING DATASETS

To assess the generalization ability of our approach across different data domains, we evaluate CROp in a multi-dataset training setup. Specifically, we combine the XGaze and EVE datasets, which are similar in size, and train a single model using their union. As shown in Table 10, this training setup preserves the accuracy of the best single-dataset model (trained on XGaze) on EyeDiap, while also improving performance on MPIIFaceGaze. This suggests that our formulation remains effective even when aggregating data from heterogeneous sources, without requiring normalization to align datasets.

## E    CROP IMPLEMENTATION

The core of our CROp approach lies in estimating the rotation that maps between the coordinate systems of the camera and the face crop, formulated as a constrained optimization problem. This can be implemented in just a few lines of code, assuming access to a library implementation of the Kabsch algorithm Kabsch (1978); Umeyama (1991). Listing 1 shows our Python implementation.

## F    USE OF LARGE LANGUAGE MODELS

Large Language Models (LLMs) were used as an assistive tool to revise the writing of this manuscript (*e.g.*, grammar, phrasing). The research ideas, experiments, and conclusions are entirely our own, and the authors take full responsibility for the scientific content.

Listing 1: Main function of CROp.

```python
import numpy
from scipy.spatial.transform import Rotation

def get_CROp_rotation(bbox, cam_matrix):
    # Compute the optimal rotation aligning crop rays to camera rays
    x1, y1, x2, y2 = bbox
    crop_center = numpy.array([(x1 + x2) / 2.0, (y1 + y2) / 2.0])

    points = numpy.array([
        [crop_center[0], crop_center[1], 1],
        [x1, y1, 1],
        [x2, y1, 1],
        [x1, y2, 1],
        [x2, y2, 1]]).T

    ray_cam = numpy.linalg.inv(cam_matrix) @ points
    ray_cam /= numpy.linalg.norm(ray_cam, axis=0)

    crop_matrix = cam_matrix.copy()
    crop_matrix[:2, 2] = crop_center
    ray_crop = numpy.linalg.inv(crop_matrix) @ points
    ray_crop /= numpy.linalg.norm(ray_crop, axis=0)

    # Solve using Kabsch algorithm
    R_opt, err = Rotation.align_vectors(ray_crop.T, ray_cam.T, weights=[
        numpy.inf, 1, 1, 1, 1])
    return R_opt.as_matrix(), err
```