# OpenReview forum: "Normalization vs. Crops: Learning Gaze Representations via Constrained Rotation Optimization"
_ICLR.cc/2026/Conference — ICLR 2026 Conference Withdrawn Submission_

### Official Review · Reviewer_pgzd · 2025-10-27

**Soundness:** 2
**Presentation:** 2
**Contribution:** 2
**Rating:** 2
**Confidence:** 4

**Summary:**

This paper formalizes a normalization-free framework for gaze estimation through a constrained rotation optimization that treats cropping as a virtual camera rotation, combined with a joint gaze–head pose learning model. The idea is theoretically clean and the empirical evaluation is thorough. However, the actual contribution is incremental, as the proposed method’s advantage over standard normalization pipelines is relatively small and mostly in noisy or extreme pose conditions. The claim of providing a “reliable alternative” is only partially supported, since the improvement margins are minor and the reliance on detector stability remains.

**Strengths:**

1. The paper tackles a well-defined and practical problemL: the trade-off between normalization-based and crop-based gaze estimation.

2. The multi-task design that jointly predicts gaze and head pose is technically sound and demonstrates robustness under real-world degradations

3. The experimental analysis is comprehensive, covering multiple datasets, realistic pre-processing conditions, and various degradation scenarios, which supports the paper’s claims.

4. The method provides measurable efficiency benefits

**Weaknesses:**

1. Despite the theoretical novelty, the practical performance improvements over standard normalization are modest (typically within 5–10%), and not statistically or conceptually transformative.

2. The method’s dependence on accurate bounding boxes introduces a hidden assumption of reliable face detection, which undermines the “normalization-free” claim.

3. The mathematical formulation is overcomplicated relative to its empirical gain; the optimization step essentially reparameterizes standard coordinate alignment.

4. There is a lack of comparison with stronger modern baselines (e.g., transformer-based gaze estimation, or geometry-aware self-supervised methods), which makes the claimed “reliability” less convincing.

5. Writing-wise, the paper spends too much space re-deriving basic camera geometry and too little analyzing failure cases or uncertainty in predictions.

**Questions:**

1. How sensitive is the proposed CROp formulation to detection noise or imperfect face bounding boxes? Since the method claims to be “normalization-free,” it would be helpful to quantify how robust it remains when the detector produces slight misalignments.

2. Could the authors clarify why the joint gaze–head pose model cannot be directly trained under a standard coordinate alignment or transformer-based architecture? In other words, what concrete advantage does CROp offer beyond reparameterization?

---

### Official Review · Reviewer_sLFB · 2025-10-30

**Soundness:** 3
**Presentation:** 3
**Contribution:** 2
**Rating:** 4
**Confidence:** 3

**Summary:**

In the context of appearance-based gaze estimation this paper approximates an image crop by a rotation, allowing the cropped image to be explained with a rotation of the original camera. The authors combine this with multi-task learning to jointly predict gaze and head pose. They demonstrate that crop-based estimation can match or exceed normalisation-based methods, particularly under extreme head poses and noisy preprocessing.

**Strengths:**

The mathematical formulation of CROp is rigorous and well-motivated. The constrained optimisation problem (Eq. P) elegantly captures the relationship between crop and camera coordinates, with the Kabsch algorithm providing a principled solution.

The paper includes extensive experiments across multiple datasets (MPIIFaceGaze, ETH-XGaze, EYEDIAP, EVE) with both cross-dataset and within-dataset evaluations.

The approach eliminates landmark detection, reducing computational cost by 23% while maintaining or improving accuracy. This is significant for real-world deployment.

The paper systematically examines the contribution of each component (CROp transformation vs. multi-task learning) and tests robustness under synthetic noise and occlusions.

**Weaknesses:**

While the formalization is clean, the core ideas are not entirely new:
- Using face crops for gaze estimation has been explored (Gaze360, GazeOnce)
- Multi-task learning for gaze and head pose has been proposed before
- The main contribution is formalising the geometric relationship, which, while useful, feels incremental

The improvements are dataset-dependent and sometimes marginal:
- On XGaze (Table 1), CROp GT achieves 7.41° vs. Norm 7.19° (worse)
- Improvements are most pronounced on smaller/noisier datasets (MPII, EYEDIAP)
- Within-dataset results (Table 2) show mixed outcomes in GT setting

The paper primarily compares against normalization but doesn't adequately compare with other normalization-free approaches:
- EFE (Balim et al., 2023) is mentioned but not empirically compared
- Gaze360's coordinate system is discussed but not experimentally evaluated
- Missing comparisons with other recent methods (e.g., the CLIP-based methods cited)

The paper acknowledges that "no single rotation can perfectly align the two views" but doesn't quantify how large this approximation error is.

Figure 5 shows degradation at extreme off-axis angles, but the analysis is limited.

The assumption that gaze origin coincides with bounding box centre is mentioned in the conclusion but not analysed.

Table 4 shows that adding head pose sometimes hurts gaze performance. The paper dismisses this as "likely due to the model's balancing" without deeper investigation. No analysis of how to weight the two losses optimally.

**Questions:**

1. Can you provide the actual approximation error of the CROp transformation (i.e., the residual from the constrained optimization)?
2. How sensitive is the method to bounding box quality? The ±30% threshold in Table 5 seems high - what happens in more realistic scenarios?
3. Why not compare directly with Gaze360's coordinate system approach experimentally?
4. Can you provide computational cost comparisons for training time, not just inference?
5. How does performance vary with face size in the image?

---

### Official Review · Reviewer_91af · 2025-11-01

**Soundness:** 3
**Presentation:** 3
**Contribution:** 3
**Rating:** 6
**Confidence:** 5

**Summary:**

This paper proposes CROp (Constrained Rotation Optimization), a novel gaze estimation method that eliminates the need for face normalization, landmark detection, and head pose estimation by modeling face cropping as a virtual camera rotation. It uses multi-task learning to jointly estimate gaze and head pose, improving robustness in unconstrained environments. Experiments across multiple datasets show that CROp outperforms traditional normalization methods, especially under extreme head poses and noisy data, while reducing computational cost.

**Strengths:**

1. The paper is well-written and easy to follow.

2. The proposed  CROp method offers an effective pre-processing approach for gaze estimation, eliminating the need for precise head pose estimation and facial landmark detection as those in conventional normalization.

3. Experiments are extensive, including both within- and cross-dataset evaluation, estimation errors, computational cost, influences of boundbox noise and occlusions.

**Weaknesses:**

1. While the paper demonstrates that joint estimation of gaze and head pose improves robustness, the necessity of incorporating head pose as a joint task with gaze estimation is not fully justified. It would be helpful to provide a more detailed explanation of why head pose estimation is critical for improving gaze accuracy, especially in unconstrained settings. The paper could also address potential trade-offs or situations where gaze estimation might still perform well without explicitly estimating head pose.

2. The process of face cropping is not clearly detailed in the paper. Specifically, it is important to clarify how the crop is performed, including the following aspects:
    Crop Size: What is the size of the cropped region? Does the crop size vary depending on the subject or image? Does it adapt to different head poses or facial feature sizes?
     Facial Feature Position Constraints: After cropping, are there any constraints on the positions of facial features (such as the eyes, nose, and mouth) within the cropped region? For example, are the facial features aligned or normalized in any way before proceeding with gaze estimation?

3. While CROp generally outperforms the normalization method (Norm) in most cases, there are instances where Norm performs better, particularly in cross-dataset experiments that are trained on the XGaze dataset. It would be useful to further analyze and discuss these cases, particularly the conditions under which Norm may outperform CROp.

**Questions:**

The paper primarily uses ResNet18 as the backbone for the experiments. While this is a reasonable choice, it would be beneficial to test CROp with other popular backbones and state-of-the-art (SOTA) methods to assess its generalizability and effectiveness across different architectures.

---

### Note · Authors · 2025-11-13

I have read and agree with the venue's withdrawal policy on behalf of myself and my co-authors.